# Economic Connectiveness and Pro-Poor Growth in Sub-Saharan Africa: The Role of Agriculture

Maria Sassi 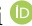

Department of Economics and Management, University of Pavia, 27100 Pavia, Italy; maria.sassi@unipv.it

**Abstract:** In Sub-Saharan Africa, economic growth is essential for poverty reduction, and pro-poor growth is the renewed focus of today's political debate. The present paper adds to the literature on the growth–inequality relationship. It provides an in-depth analysis of the potential role of agriculture in promoting pro-poor growth in rural and urban areas compared with that of other activities. This aspect still lacks rigorous empirical support. Using the Nexus project SAMs by the International Food Policy Research Institute, this study identifies the level of 'keyness' of 36 activities (12 are agricultural) in nine Eastern, Western, and Central African countries using the inter-industry linkages analysis. Afterwards, it investigates the income distribution multipliers effects of activities growth across households classified in quantiles in rural and urban areas. Therefore, the paper adds to the literature, mainly focused on rural poverty and information on the growth effect on urban poverty, which is important in the context of rapid urbanization and the growing number of poor people in African cities. Apart from country-specific factors, the results confirm the strong integration of agriculture with the economy. The growth of key agricultural activities presents the most pronounced multiplicative effect on the income of rural households in the lowest quantiles. Poor urban households also benefit from their growth, but not to the same extent as rural households with an increase in the rural–urban income gap.

**Keywords:** pro-poor growth; rural–urban poverty; agricultural development; multiplier analysis; Sub-Saharan Africa

## 1. Introduction

COVID-19, conflict and climate change are recent global shocks that have had a tremendous human cost to developing countries as hundreds of millions of people are falling into poverty [1]. Under this scenario, sub-Saharan Africa is the area of greatest concern. Many countries in Africa have the highest poverty rates in the world. Furthermore, recent projections point to poverty, particularly extreme poverty, becoming a predominant African phenomenon [2].

The literature provides clear evidence that economic growth is an essential requirement and often the primary contributor to poverty reduction [3,4]. Given the current economic recession and rising poverty rates across the African continent, there is renewed interest in the pro-poor growth process. Consequently, today's political debate concerns accelerating the growth and participation of the poor in this process.

In this debate, there is also a renewed focus on the potential role of pro-poor growth agricultural development, especially in rural areas where the majority of poor households are located [5–8]. This role has been long neglected [6]. However, the recent literature acknowledges the positive relationship between poverty reduction and increasing agricultural growth rates in Sub-Saharan Africa [1,9]. This relationship is ascribed to the extensive participation of the poor in agriculture, a sector that still provides a relatively significant contribution to the economy, particularly if its indirect impact on the growth of other sectors is accounted for [10].

This paper contributes to this debate and adds to the literature on the growth–inequality relationship an in-depth analysis of the role of the agricultural sector [11]. Most empirical literature uses cross-sectional data to verify this relationship [12,13]. The current study complements this perspective using a different approach. Focusing on nine East, Western and Central African countries, it first identified the key economic sectors with strong potential to drive their domestic economic growth. Afterwards, it studied the magnitude of effects produced by an exogenous demand injection of one unit in the individual 36 activities in the dataset on households classified according to income quantiles both in rural and urban areas. The literature argues that raising agricultural production and productivity in developing countries reduces poverty more than comparable growth in other sectors, specifically focusing on rural areas. However, this assertion has rarely been based on evidence from rigorous empirical investigations [14]. In this respect, the research question of this paper is to understand if a stimulus to the demand in the key economic sectors promotes a distributional effect that benefits more low-income households in rural and urban areas than the better-off distinguishing among economic activities. The current analysis is, therefore, grounded in a relative approach to pro-poor growth [15].

The research question of the current study is of specific importance when related to the potential role of the agricultural sector in pro-poor growth. In fact, despite the recognition of the relevance of the sector for growth and poverty reduction, public, private, and foreign investment in agriculture remains weak and inadequate in Africa [16]. More precisely, policymakers were discouraged by the uncertainties about the effectiveness of these investments. This study can assist them at least in two ways: correctly identifying key economic sectors with the important potential to stimulate economic growth domestically and revealing whether a clear pattern emerges between the key sectors and pro-poor growth.

The distinction between rural and urban households made by the present study is crucial when analyzing the potential pro-poor growth of economic and agricultural activities. As previously highlighted, the available studies on this aspect mainly focus on poverty in rural areas. However, in the current context of rapid urbanization, Africa is experimenting with a growing number of poor people in cities because of the recent global downturn and lack of employment and income [17,18]. Therefore, pro-poor growth strategies should account for the different geographic locations of poverty. Moreover, many people living in poverty in urban areas have strong links to rural areas through the agricultural sector. These connections and their effects on urban poverty reduction still deserve a clear understanding.

Since Hirshman's findings [19], sectoral interdependency has been considered a crucial source of economic development for a country [20]. The literature underlines the positive relationship between the extent of intra-industry linkages and the level of production of an economy, which is a measure of economic growth [21]. For this reason, input–output-based linkage analysis techniques have been used extensively to identify the leading sectors in an economy and to formulate development strategies [22–32].

The analysis of key links is a rigorous economic approach which makes it possible to identify quantitatively the interdependencies between the various economic sectors and to estimate the wider systemic impacts.

This research followed this approach to compare the 'keyness' of a sector with other sectors of the economy. However, instead of using input–output matrices widely adopted in the literature, Social Accounting Matrices (SAM) were employed to account for both production and consumption linkages. Sectors with above-average linkages promote economic development and structural change. They act to speed up and amplify initial minor changes and eventually affect the entire economy [33].

Following Cohen [34], the paper adopted the Relative Distributive Measure (RDM) using the income multipliers from the SAM to analyze the distribution multiplier effect across households and the underlying structural bias. As income by household groups is a more relevant indicator of social welfare than gross output, it should complement the

information on the role of sectors in development based on the analysis of inter-industry linkages. This consideration is strengthened by the fact that household income is also closer to the efficiency notion of value added than gross output [35].

The current paper applied the above-described methodology in the Democratic Republic of Congo (DRC), Ethiopia, Kenya, Mali, Niger, Nigeria, Senegal, Tanzania, and Zambia using the most recent comparable SAMs produced by the International Food Policy Research Institute (IFPRI) and referred to 2018 and 2019.

The comparison between countries has allowed us to understand better the most crucial aspects of the pro-poor growth process in Africa. In addition, the literature provides only three studies using a similar approach, but none focused on African countries. In fact, they refer to the Netherlands [34], Russia and China [35], and Nepal [36]. However, these papers propose an analysis of the distribution of the multiplier effects of injection in sectors on the sectors themselves and households' income to investigate the respective structural bias. As a result, the identification of key sectors is based only on a demand-driven model. The current paper adds the supply side dimension to this identification process considering both backward and forward linkages.

The remainder of the paper is structured as follows. Section 2 describes the method and data and methods used in the employed in the econometric analysis, Section 3 presents and discusses results, and Section 4 concludes.

## 2. Data and Methods

The paper referred to the latest SAMs published by IFPRI for Sub-Saharan African countries as part of the Nexus Project. This project has established common data standards, procedures, and classification systems used to construct and update the national SAMs, allowing for a more robust comparison of the economic structure across countries (https://www.ifpri.org/project/nexus-project accessed on 4 January 2023). In fact, the size of multipliers differs to a certain extent according to the level of account aggregation [35]. The current study used the freely available online SAMs of Ethiopia (2018), Kenya (2019), Tanzania (2018), and Zambia (2019) in Eastern Africa, the DRC (2018) in Middle Africa, and Mali (2018), Niger (2018), Nigeria (2018), and Senegal (2018) in Western Africa [37–45].

A SAM is a square matrix based on the double accounting principle, whereby each account represents the revenues and payments recorded in rows and columns, respectively. The SAMs produced by the Nexus Project include Activities, Commodities, Factors, Enterprises, Households, Government, Taxes, Investment, and Rest of the World accounts.

The current study focuses specifically on activities and households. The standard Nexus SAMs disaggregate activities in the 42 sectors shown in Table 1 with the codes used in the paper. Each refers to a set of industries classified according to the 4-digit International Standard Industrial Classification system (ISIC Revision 4), with the agricultural activities further disaggregated according to the FAO's categorization system [46].

Not all activities are represented in all the countries under study. They refer to the agrifood system and reflect the specific country's production capacity. The absent sectors are in the category of agricultural commodities and livestock and fisheries. For a cross-country comparison, the current study merged sugarcane, tobacco, cotton and fibers, and coffee, tea and cocoa in one sector (suco) and cattle and raw milk, poultry and eggs, other livestock, and fisheries in another sector (lifi). The paper maintained the detailed original classification for the other accounts.

For households, a Nexus SAM separates national populations in rural (RU) and urban (UR) using the official definition of these areas adopted at the country level. A SAM includes five representative household groups for the population in each of these two areas. These categories represent the income quintiles defined at the national level. Therefore, rural and urban quantiles are comparable at the country level. Moreover, each of them represents one-fifth of the national population.

**Table 1.** Standard Nexus SAM activities and codes.

| Code | Description | Code | Description |
|---|---|---|---|
| maiz | Maize | text | Textiles, clothing and footwear |
| rice | Rice | wood | Wood and paper products |
| ocer | Other cereals | chem | Chemicals and petroleum |
| puls | Pulses | nmet | Non-metal minerals |
| oils | Oilseeds | metl | Metals and metal products |
| root | Roots | mach | Machinery, equipment and vehicles |
| vege | Vegetables | oman | Other manufacturing |
| suco | Sugarcane (sugr) | elec | Electricity, gas and steam |
| | Tobacco (toba) | watr | Water supply and sewage |
| | Cotton and fibers (cott) | cons | Construction |
| | Coffee, tea and cocoa (coff) | trad | Wholesale and retail trade |
| frui | Fruits and nuts | tran | Transportation and storage |
| ocrp | Other crops | hotl | Accommodation and food services |
| lifi | Cattle and raw milk (catt) | comm | Information and communication |
| | Poultry and eggs (poul) | fsrv | Finance and insurance |
| | Other livestock (oliv) | real | Real estate activities |
| | Fisheries (fish) | bsrv | Business services |
| fore | Forestry | padm | Public administration |
| mine | Mining | educ | Education |
| food | Processed foods | heal | Health and social work |
| beve | Beverage and tobacco | osrv | Other services |

( . . . ) code in the original SAM. Source: author's elaboration from [37–45].

In all the countries analyzed, the majority of the population lives in rural areas where the poor are concentrated (Table 2).

**Table 2.** Percentage of population by area, wealth category, and country.

| | DRC | Ethiopia | Kenya | Tanzania | Zambia | Mali | Niger | Nigeria | Senegal |
|---|---|---|---|---|---|---|---|---|---|
| RU_1 | 16.2 | 19.4 | 17.6 | 17.8 | 18.2 | 18.9 | 19.9 | 17.6 | 17.7 |
| RU_2 | 13.7 | 19.1 | 16.1 | 17.0 | 16.8 | 18.1 | 19.6 | 15.6 | 16.4 |
| RU_3 | 12.6 | 18.5 | 14.3 | 15.8 | 13.0 | 16.7 | 18.1 | 12.9 | 12.7 |
| RU_4 | 10.5 | 17.0 | 11.0 | 13.1 | 7.4 | 13.6 | 15.9 | 9.9 | 7.1 |
| RU_5 | 8.2 | 11.6 | 5.2 | 7.3 | 2.9 | 7.1 | 10.4 | 7.3 | 2.2 |
| RURAL | **61.2** | **85.7** | **64.2** | **71.2** | **58.3** | **74.3** | **83.8** | **63.4** | **56.2** |
| UR_1 | 3.8 | 0.6 | 2.4 | 2.2 | 1.8 | 1.2 | 0.1 | 2.4 | 2.3 |
| UR_2 | 6.3 | 0.9 | 3.9 | 3.0 | 3.2 | 1.9 | 0.5 | 4.4 | 3.6 |
| UR_3 | 7.4 | 1.4 | 5.7 | 4.1 | 7.0 | 3.3 | 1.9 | 7.0 | 7.3 |
| UR_4 | 9.5 | 3.0 | 9.0 | 6.9 | 12.6 | 6.4 | 4.1 | 10.1 | 12.9 |
| UR_5 | 11.8 | 8.4 | 2.4 | 12.7 | 17.1 | 12.9 | 9.6 | 12.6 | 17.8 |
| URBAN | **38.8** | **14.3** | **35.8** | **28.8** | **41.7** | **25.7** | **16.2** | **36.6** | **43.8** |
| Total | 100.0 | 100.0 | 100.0 | 100.0 | 100.0 | 100.0 | 100.0 | 100.0 | 100.0 |

Source: [37–45].

### 2.1. Economic Linkages

The paper applied a demand- and supply-side model to the SAM multipliers to compute the backward and forward linkages in the analyzed countries.

Backward linkages capture the relationship of a particular sector with those upstream from which it purchases inputs [47]. A coefficient matrix (A) was constructed to obtain them by dividing each column's account in the SAM by its total. The Government, Taxes, Investment, and Rest of the World accounts were excluded from this computation. Following the standard convention, the current study treated these accounts as exogenous by assuming that their expenditures are independent of income [48]. The literature supports this choice in competitive economies because it reflects the fact that government expenditures are usually policy-determined, the Rest of the World is not controlled by the domestic economy, and investment is exogenously determined due to the static nature of the SAM [49]. The analysis treats the remaining accounts as endogenous. Interpretation of the results should consider that the size of the multipliers depends, in part, on the selected exogenous variables [35].

According to the principle of double-entry accounting underlining the SAMs, a whole economy can be represented in a linear form. Denoting the endogenous variables by $y$ and the exogenous variables by $x$, total demand is given by the following equation:

$$y = Ay + x = (I - A)^{-1}x = Mx \tag{1}$$

where $A$ is the matrix of the average propensities obtained by dividing each flow in the SAM by the respective column total, and $M$ is the aggregate multiplier matrix or input inverse matrix given by the identity matrix ($I$) minus $A$.

The paper used the industry-by-industry multipliers of the $M$ matrix, corresponding to the Leontief inverse matrix, to compute the backward linkages.

If sector $j$ increases by one unit, its output, backward linkages ($B$) measure the increase in demand for inputs from sector $j$ to upstream sectors $i$. The backward linkages of each activity were computed as the column sum of the elements $m_{ij}$ of the Leontief inverse matrix as follows:

$$B_j = \sum_{i=1}^{n} m_{ij} \quad i = 1, \ldots, n \tag{2}$$

The concept of sector "keyness" is a relative notion that depends on other sectors. Therefore, following the work of Rasmussen [50] and Hirschman [19], Equation (2) was normalized to obtain an indicator of backward linkages (BL) by dividing the backward linkage of sector $j$ by the simple average of all backward linkages. This indicator is called "power of dispersion" and is formulated as below:

$$BL_j = \frac{B_{.j}}{\frac{1}{n}\sum_{j=1}^{n} B_{.j}} j = 1, \ldots, n \tag{3}$$

If $BL_j$ is greater (lower) than 1, a unit change in final demand in sector $j$ will generate an increase in the activity of the economy above (below) the average. Hence, the sector draws heavily (slightly) from the rest of the economy.

Forward linkages account for the interconnections of a sector with those downstream to which it sells its output [47]. Following Beyers [26] and Jones [51], the current study used the Ghosh [52] model for their computation instead of the Leontief inverse matrix. The literature shows that the calculation of forward linkages as a row total of the Leontief inverse matrix has a hypothetical and no general economic interpretation [33]. At the increase in output of sector $j$, forward linkages à la Leontief assume a simultaneous increase in the final demand of every supplied activity. Ghosh [52] argues that a similar model is only appropriate when different sectors of a country's economy are under monopolistic control, and all but one resource is scarce. The countries of the sample do not exhibit these characteristics.

Therefore, to compute forward linkages, each SAM's row account was divided by the row sum, obtaining a matrix of allocation coefficients (*C*) from which we constructed the Ghosh output inverse matrix (*G*) as follows:

$$G = (I - C)^{-1} \tag{4}$$

In other words, the value of the total intermediate sales of sector *i* was calculated as a proportion of the value of the sector *i*'s total output [47].

The row sums of the elements of the Ghosh-inverse matrix ($g_{ij}$), called allocation coefficients, provide the total forward linkages (*F*). These are computed as:

$$F_i = \sum_{j=1}^{n} g_{ij} \quad j = 1, \dots, n \tag{5}$$

Following Oosterhaven [53–56], the impact of a one-unit change in input from sector *i* on the price of total inputs of all sectors was measured.

The paper used the normalization method introduced above to compute an indicator of forward linkages (*FL*), also called the "sensitivity of dispersion", as:

$$FL_i = \frac{F_{i.}}{\frac{1}{n} \sum_{i=1}^{n} F_{i.}} \quad i, j = 1, \dots, n \tag{6}$$

A value of $FL_i$ larger (lower) than 1 indicates that a unit change in the final demand of all sectors would create an above- (below-) average production increase in sector *i*. The sector is a key (minor) supplier for the rest of the economy.

The economic sectors were grouped into four categories by comparing the value of backward and forward linkages. The paper defined as key sectors (*K*) those with both *BL* and *FL* above one, forward linkage-oriented sectors, (*F*) those with only *FL* greater than one, backward linkages-oriented sectors, (*B*) those with only *BL* greater than one, independent or weak linkages-dependent sectors, and (*N*) those with both *BL* and *FL* lower than one.

*BLs* and *FLs* were computed using both inter-and intra-sectoral trade. In developing countries, transactions within a sector (i.e., intra-sectoral transactions) often constitute an essential element of value in a sector's value chain. By including this component in the current analysis, the importance of this portion of the trade is accounted for.

Following Hazari [22], the paper computed the coefficient of variation index (*CV*) of the *BLs* and FLs to detect the possible influence of a few sectors. For example, let us assume that sector *j* purchases an extremely large amount of inputs from one or two other sectors. In this case, the *BL* of sector *j* is greater than those of the other sectors, but its potential to spread growth impulses throughout the economy is limited to a low number of sectors. The *CVj* and *CVi* were computed using the sectoral elements of *Bj* and *Fi*, respectively, as the standard deviation divided by the mean value. A high (low) value of *CVj* implies that sector *j* purchases inputs from a few sectors (most of the sectors equally). Similarly, a high (low) *CVi* indicates that sector *i* sells its good or services to a few sectors (most sectors). The current study uses the average *CV* value across sectors to highlight those with the highest dispersion around the mean value.

The analysis also includes the Spearman [57] correlation coefficient that is adopted to verify the strength and direction of the association of *FL*, *BL*, and the level of "keynes" of the sectors between countries. The paper considers the association significant when the *p*-value of the two-tailed statistical significance was less than 0.05. Table 3 shows the guide followed to describe the strength of the correlation.

**Table 3.** Strength of correlation categories.

| Absolute Value of Spearman Correlation Coefficient | Strength of the Correlation |
| :---: | :---: |
| 0.00–0.19 | Very weak |
| 0.20–0.39 | Weak |
| 0.40–0.59 | Moderate |
| 0.60–0.79 | Strong |
| 0.80–1.0 | Very strong |

*2.2. Relative Distributive Measure*

The current study followed Cohen [34] by using the Relative Distributive Measure (*RDM*) to study the distribution of multiplier effects on households (h) of an exogenous demand injection in the activities *j* and assess the underlying structural bias in the country analyzed. This injection can be from investment, government spending or exports.

Following Cohen [35], the *RDM* was computed as follows:

$$RDM_{hj} = \frac{m_{hj} / \sum_{h=1}^{m} m_{hj}}{Y_{h0} / \sum_{j=1}^{n} Y_{h0}} \quad h = 1, \dots, m; j = 1, \dots, n \tag{7}$$

where $m_{hj}$ is the income multiplier of household category *h* generated from a demand injection in activity *j*. It is part of the block of the *M* matrix defined by Equation (1). The denominator is the income share of household group *h* at year 0. It is from the values found in the SAMs for the base year. *RDM* takes values equal to one, greater than one and lower than one for neutral, positive, and negative distributive effects, respectively. In other words, a value of *RDM* equal to 1 means that the sectoral transfer would reproduce the same sectoral distribution pattern of the base year. A value above (below) unity for a household group would mean an increase (deterioration) in its income share relative to the base year and, therefore, a positive (negative) income growth bias towards that household group.

## 3. Results and Discussion

The comparison of the sectors ranked according to their "keyness" shows a relative dispersion across countries, with some exceptions. Focusing on the key sectors, maize, pulses, oilseeds, root, vegetables, and fruits and nuts are in this category in all the countries analyzed (Table 4). Moreover, agricultural activities make up the majority of the key sectors in all the countries considered. Their prevalence ranges from 50% in Niger to 69% in Nigeria. Another common feature of the key sectors category across countries is the presence of manufacturing and processing activities directly related to agriculture, among which the processed foods industry (food) is the most recurrent. This evidence confirms the potential role of agriculture in creating a value chain that can be strategic for growth.

The results also showed that many key agricultural activities populate the top ten sectors regarding the value of *BL* and especially *FL* indices (Tables 5 and 6). Therefore, they represent an important motor of economic development. In these sectors, the above-average *FLs* imply that promoting or disadvantaging the key agricultural activities will affect the factor costs of the downstream industries more than in other sectors. Consistent with the literature [4], also in the countries analyzed, the lower *BL* indices than *FL* indices are due, at least in part, to the fact that agriculture is still labor-intensive. A common feature across the analyzed countries is that the majority of farmers are smallholders practicing low productivity, low technology, and high labor-intensity agriculture [58–60]. For example, they contribute to 75 percent of the total agricultural output of Kenya, 90 percent of Ethiopia, and 99 percent of Nigeria.

**Table 4.** Key sectors and respective CVi, CVj.

| DRC | Ethiopia | Kenya | Tanzania | Zambia | Mali | Niger | Nigeria | Senegal |
|---|---|---|---|---|---|---|---|---|
| [2.460; 6.319] | [6.222; 2.312] | [2.038; 2.032] | [2.204; 2.144] | [3.089; 2.839] | [2.513; 2.341] | [1.986; 2.220] | [1.864; 2.009] | [2.372; 2.423] |
| maiz (2.619; 6.108) | maiz (6.201; 1.799) | maiz (2.030; 1.587) | maiz (2.232; 1.684) | maiz (2.842; 2.062) | maiz (2.476; 1.749) | maiz (1.759; 1.422) | maiz (1.824; 1.489) | maiz (2.172; 1.751) |
| rice (2.817; 6.162) | rice (6.080; 1.722) | rice (1.839; 1.568) | rice (2.234; 1.726) | rice (2.978; 2.115) | rice (2.605; 1.953) | ocer (2.430; 1.973) | ocer (1.765; 1.387) | ocer (2.160; 1.757) |
| ocer (2.385; 6.115) | ocer (6.286; 2.007) | ocer (1.885; 1.425) | puls (2.305–1.810) | ocer (2.937; 2.215) | ocer (2.419; 1.760) | puls (1.984; 1.672) | puls (2.496; 2.139) | puls (2.480; 1.858) |
| puls (2.483; 6.068) | puls (6.152; 1.998) | puls (2.150; 1.559) | oils (2.246; 1.753) | puls (2.933; 2.194) | puls (2.713; 1.902) | oils (2.012; 1.615) | oils (2.059; 1.810) | oils (2.598; 2.231) |
| oils (2.498; 6.003) | oils (6.087; 1.690) | oils (1.867; 1.388) | root (2.302; 1.717) | oils (2.985; 2.232) | oils (2.627; 1.934) | root (2.048; 1.806) | root (2.029; 1.743) | root (2.366; 1.864) |
| root (2.575; 6.034) | root (6.131; 1.782) | root (2.074; 1.487) | vege (2.177; 1.613) | root (2.977; 2.150) | root (2.567; 1.888) | ege (2.146; 1.837) | vege (1.962; 1.663) | vege (2.522; 2.088) |
| vege (2.452; 5.992) | vege (6.109; 1.781) | vege (2.220; 1.620) | frui (2.228; 1.816) | vege (2.951; 2.066) | vege (2.810; 2.001) | suco (1.975; 1.709) | frui (1.719; 1.425) | frui (2.398; 2.017) |
| frui (2.434; 5.989) | frui (6.102; 1.732) | frui (1.997; 1.554) | food (2.174; 1.868) | frui (2.971; 2.103) | frui (2.524; 1.887) | frui (2.127; 1.832) | ocrp (1.580; 1.367) | text (2.459; 1.989) |
| ocrp (2.390; 5.978) | ocrp (6.095; 1.737) | food (2.203; 1.729) | oman (2.028; 1.647) | food (3.100; 2.470) | food (2.550; 2.065) | text (2.067; 1.941) | fore (1.917; 1.577) | oman (2.762; 2.197) |
| food (3.006; 6.217) | lifi (6.193; 2.112) | chem (1.855; 1.411) | elec (2.711; 2.035) | text (2.927; 2.164) | elec (3.323; 2.575) | chem (2.094; 1.632) | beve (1.615; 1.423) | watr (2.006; 1.732) |
| beve (2.338; 5.993) | fore (6.181; 1.839) | nmet (2.147; 1.948) | tran (2.393; 2.026) | wood (2.821; 2.442) | fsrv (2.202; 1.680) | oman (2.177; 2.201) | text (2.661; 2.195) | comm (2.613; 2.017) |
| chem (2.380; 6.035) | food (6.208; 1.949) | trad (2.134; 1.652) | fsrv (1.982; 1.919) | chem (3.041; 2.423) | real (2.291; 1.716) | elec (1.870; 1.565) | chem (1.658; 1.521) | fsrv (2.165; 1.743) |
| oman (2.217; 6.016) | comm (6.139; 1.942) | | bsrv (2.309; 2.016) | oman (2.836; 2.130) | | tran (1.970; 1.772) | fsrv (2.059; 1.782) | |
| osrv (2.309; 5.975) | real (6.200; 1.976) | | | elec (3.393; 2.289) | | hotl (1.816; 1.788) | | |
| | bsrv (6.099; 1.831) | | | watr (2.667; 2.580) | | comm (1.960; 1.749) | | |
| | osrv (6.168; 1.882) | | | | | osrv (1.808; 1.600) | | |

[ … ] Average CV computed in all 36 activities. ( … ) CV of the specific activity.

The average *CV* of the power of dispersion ranges between 1.864 in Nigeria and 6.222 in Ethiopia, while the *CV* for the sensitivity of dispersion is between 2.009 in Nigeria and 6.319 in the DRC. Focusing on the key sectors, the *CV* indicates their potential to spread growth impulses to a relatively higher number of upstream sectors than the overall sectors' average and their backstream activities (Table 4). In addition, the fact that, among the key sectors, there are industries directly related to the primary sector with a *CV* below average for the *FL* indices suggests the presence of a diversification process within the agrifood chain towards higher-value-added goods.

The *CV* computed on the *BL* indices highlights the relatively limited integration of the agricultural sectors with the downstream sectors. This result was expected. As highlighted by Sheahan and Barret [61], despite the signs of progress in the use of modern input in Sub-Saharan Africa and the specificities within countries, the adoption of inputs that include improved technologies, such as improved seed, fertilizers and other agro-chemicals, machinery, and irrigation is still low. In fact, in the countries analyzed, agriculture relies on extensive land use practices, and access to inputs and finance is still problematic. Because of this situation, agricultural productivity is lower than in other sectors, as shown in Table 7. In fact, the lower agriculture's employment share compared with the share of agricultural

value added over Gross Domestic Product (GDP) indicates that agricultural output per worker is lower than labor productivity in other sectors.

**Table 5.** Activities with *BL* above average and their respective *BL* value.

| DRC | Ethiopia | Kenya | Tanzania | Zambia | Mali | Niger | Nigeria | Senegal |
|---|---|---|---|---|---|---|---|---|
| food 1.158 | lifi 1.150 | hotl 1.086 | hotl 1.132 | cons 1.124 | elec 1.265 | oman 1.178 | puls 1.332 | oman 1.125 |
| root 1.154 | Real 1.148 | food 1.078 | food 1.130 | watr 1.121 | bsrv 1.119 | chem 1.123 | text 1.173 | fsrv 1.094 |
| frui 1.096 | root 1.108 | maiz 1.077 | root 1.079 | wood 1.115 | metl 1.116 | root 1.116 | oils 1.165 | root 1.078 |
| oils 1.095 | ocer 1.108 | root 1.077 | bsrv 1.074 | osrv 1.113 | hotl 1.087 | frui 1.107 | maiz 1.113 | hotl 1.076 |
| maiz 1.088 | puls 1.107 | vege 1.070 | elec 1.065 | maiz 1.090 | food 1.084 | elec 1.092 | vege 1.073 | oils 1.074 |
| ocrp 1.084 | oils 1.096 | puls 1.070 | puls 1.062 | padm 1.071 | mach 1.081 | hotl 1.079 | root 1.069 | wood 1.070 |
| beve 1.083 | rice 1.096 | rice 1.067 | nmet 1.054 | oman 1.069 | oman 1.070 | comm 1.078 | chem 1.066 | puls 1.069 |
| puls 1.066 | food 1.096 | frui 1.063 | fsrv 1.050 | puls 1.065 | puls 1.061 | text 1.060 | fsrv 1.063 | nmet 1.064 |
| vege 1.066 | maiz 1.093 | oils 1.063 | vege 1.049 | root 1.056 | nmet 1.060 | cons 1.055 | hotl 1.040 | padm 1.064 |
| rice 1.047 | ocrp 1.092 | ocer 1.047 | maiz 1.047 | suco 1.054 | vege 1.049 | tran 1.053 | fore 1.025 | watr 1.058 |
| ocer 1.035 | vege 1.083 | ocrp 1.044 | oman 1.047 | hotl 1.054 | ocer 1.037 | vege 1.043 | ocer 1.020 | heal 1.047 |
| scot 1.030 | fore 1.066 | trad 1.043 | frui 1.039 | food 1.038 | oils 1.033 | ocer 1.039 | padm 1.012 | vege 1.045 |
| fsrv 1.029 | frui 1.064 | nmet 1.035 | metl 1.035 | oils 1.037 | root 1.020 | maiz 1.034 | ocrp 1.011 | ocer 1.041 |
| oman 1.022 | bsrv 1.052 | cons 1.035 | oils 1.032 | ocer 1.036 | mine 1.019 | wood 1.022 | frui 1.009 | bsrv 1.034 |
| padm 1.015 | comm 1.049 | padm 1.035 | padm 1.022 | metl 1.035 | maiz 1.019 | food 1.020 | beve 1.006 | frui 1.032 |
| chem 1.014 | hotl 1.030 | educ 1.032 | tran 1.021 | vege 1.029 | rice 1.018 | osrv 1.017 | | maiz 1.025 |
| osrv 1.013 | osrv 1.027 | wood 1.030 | ocrp 1.019 | nmet 1.022 | frui 1.011 | educ 1.015 | | text 1.022 |
| | tran 1.016 | rice 1.016 | rice 1.016 | mach 1.021 | fsrv 1.005 | real 1.013 | | cons 1.013 |
| | padm 1.011 | chem 1.013 | cons 1.002 | text 1.019 | real 1.005 | oils 1.013 | | mach 1.003 |
| | educ 1.008 | metl 1.007 | wood 1.000 | frui 1.018 | | suco 1.013 | | comm 1.001 |
| | | heal 1.001 | | chem 1.011 | | puls 1.007 | | |
| | | | | educ 1.009 | | | | |
| | | | | rice 1.003 | | | | |

**Table 6.** Activities with *FL* above average and their respective *FL* value.

| DRC | Ethiopia | Kenya | Tanzania | Zambia | Mali | Niger | Nigeria | Senegal |
|---|---|---|---|---|---|---|---|---|
| rice 1.545 | rice 1.422 | maiz 1.376 | rice 1.344 | ocer 1.426 | rice 1.340 | ocrp 1.286 | puls 1.442 | ocer 1.309 |
| maiz 1.512 | maiz 1.413 | ocer 1.355 | chem 1.338 | maiz 1.413 | elec 1.313 | maiz 1.268 | text 1.374 | maiz 1.302 |
| ocer 1.511 | ocer 1.330 | rice 1.349 | ocer 1.293 | rice 1.364 | oils 1.266 | ocer 1.266 | maiz 1.279 | oman 1.276 |
| puls 1.457 | watr 1.234 | root 1.281 | maiz 1.290 | frui 1.294 | puls 1.258 | oils 1.251 | ocer 1.256 | puls 1.266 |
| oils 1.272 | fsrv 1.223 | oils 1.280 | lifi 1.259 | vege 1.223 | ocer 1.244 | frui 1.249 | oils 1.215 | rice 1.234 |
| root 1.265 | elec 1.208 | watr 1.256 | root 1.257 | oman 1.216 | maiz 1.244 | beve 1.230 | suco 1.204 | root 1.233 |
| frui 1.262 | root 1.187 | puls 1.244 | oils 1.221 | root 1.197 | tran 1.244 | chem 1.227 | vege 1.196 | fsrv 1.220 |
| vege 1.253 | oils 1.178 | vege 1.224 | vege 1.209 | real 1.193 | suco 1.224 | fsrv 1.221 | rice 1.190 | oils 1.201 |
| text 1.243 | chem 1.164 | lifi 1.206 | elec 1.198 | fore 1.191 | vege 1.223 | comm 1.215 | fore 1.189 | elec 1.188 |
| ocrp 1.236 | beve 1.163 | oman 1.184 | fore 1.188 | lifi 1.174 | root 1.191 | root 1.208 | root 1.173 | comm 1.181 |
| food 1.213 | osrv 1.148 | food 1.183 | osrv 1.143 | text 1.159 | chem 1.164 | elec 1.193 | food 1.163 | real 1.163 |
| beve 1.207 | fore 1.145 | real 1.171 | comm 1.142 | puls 1.157 | watr 1.162 | rice 1.189 | fsrv 1.150 | tran 1.158 |
| bsrv 1.170 | food 1.143 | comm 1.164 | puls 1.142 | watr 1.151 | fsrv 1.160 | tran 1.188 | osrv 1.147 | osrv 1.147 |
| watr 1.163 | ocrp 1.134 | beve 1.149 | beve 1.139 | oils 1.146 | comm 1.147 | vege 1.186 | frui 1.144 | text 1.140 |
| trad 1.121 | lifi 1.127 | osrv 1.141 | watr 1.137 | comm 1.140 | real 1.142 | puls 1.173 | comm 1.114 | vege 1.130 |
| comm 1.119 | vege 1.127 | frui 1.132 | real 1.137 | tran 1.119 | food 1.130 | suco 1.162 | lifi 1.110 | watr 1.130 |
| real 1.115 | frui 1.124 | trad 1.130 | food 1.116 | elec 1.101 | osrv 1.117 | osrv 1.144 | watr 1.110 | suco 1.121 |
| osrv 1.099 | wood 1.114 | text 1.128 | text 1.113 | food 1.075 | beve 1.114 | lifi 1.138 | bsrv 1.104 | lifi 1.120 |
| tran 1.098 | bsrv 1.108 | chem 1.125 | trad 1.094 | wood 1.072 | frui 1.114 | oman 1.129 | ocrp 1.097 | fore 1.114 |
| chem 1.086 | real 1.085 | elec 1.106 | oman 1.084 | fsrv 1.056 | ocrp 1.095 | watr 1.111 | real 1.078 | frui 1.084 |
| elec 1.083 | comm 1.072 | nmet 1.010 | frui 1.079 | chem 1.056 | wood 1.081 | mine 1.108 | chem 1.076 | beve 1.080 |
| oman 1.016 | trad 1.054 | fsrv 1.006 | tran 1.074 | | fore 1.059 | text 1.095 | elec 1.062 | ocrp 1.065 |
| fore 1.006 | puls 1.039 | mine 1.005 | bsrv 1.059 | | | bsrv 1.085 | beve 1.059 | food 1.057 |
| | text 1.009 | | fsrv 1.026 | | | trad 1.035 | oman 1.055 | trad 1.032 |
| | | | | | | hotl 1.018 | trad 1.042 | chem 1.024 |

**Table 7.** Agricultural productivity and contribution to GDP.

|  | Share of Employment in Agriculture (%) * | Agricultural Value Added as a Share of GDP (%) ** |
|---|---|---|
| DRC (2018) | 64.8 | 20.1 |
| Ethiopia (2018) | 67.3 | 38.9 |
| Kenya (2019) | 54.3 | 37.0 |
| Mali (2018) | 63.0 | 40.7 |
| Niger (2018) | 72.9 | 38.9 |
| Nigeria (2018) | 35.5 | 21.4 |
| Senegal (2018) | 30.8 | 16.7 |
| Tanzania (2018) | 65.7 | 32.3 |
| Zambia (2019) | 49.6 | 3.5 |

Source: * https://data.worldbank.org/indicator/SL.AGR.EMPL.ZS accessed on 4 January 2023; ** [37–45].

The Spearman correlation coefficient indicates strong and statistically significant relationships in sectors' *FLs* indices between pairs of countries (Table 8). These similarities might be explained by the fact that technological factors rather than country factors determine these linkages. On the contrary, *BL* indices seem to be country specific. The Spearman correlation coefficient is strongly statistically significant in very few cases. Moreover, within countries, there is no relevant correlation between *BL* and *FL*. The only relevant exceptions are Ethiopia and Niger.

**Table 8.** Spearmen correlation coefficient of *BL* (below the diagonal) and *FL* (above the diagonal) between countries and spearman correlation coefficient between *BL* and *FL* within countries (on the diagonal—shaded in gray).

|  | DRC | ETH | KEN | MAL | NIG | NRA | SEN | TAN | ZAM |
|---|---|---|---|---|---|---|---|---|---|
| DRC | 0.537 ** | 0.693 *** | 0.770 *** | 0.711 *** | 0.686 *** | 0.776 *** | 0.697 *** | 0.709 *** | 0.746 *** |
|  |  | 0.000 | 0.000 | 0.000 | 0.000 | 0.000 | 0.000 | 0.000 | 0.000 |
| ETH | 0.393 * | 0.455 ** | 0.715 *** | 0.728 *** | 0.711 *** | 0.640 *** | 0.674 *** | 0.824 **** | 0.657 *** |
|  | 0.018 |  | 0.000 | 0.000 | 0.000 | 0.000 | 0.000 | 0.000 | 0.000 |
| KEN | 0.596 ** | 0.424 ** | 0.260 | 0.680 *** | 0.579 *** | 0.690 *** | 0.796 *** | 0.845 **** | 0.799 *** |
|  | 0.000 | 0.010 |  | 0.000 | 0.000 | 0.000 | 0.000 | 0.000 | 0.000 |
| MAL | 0.315 | 0.099 | 0.381 * | −0.022 | 0.751 *** | 0.732 *** | 0.809 **** | 0.734 *** | 0.679 *** |
|  | 0.062 | 0.564 | 0.022 |  | 0.000 | 0.000 | 0.000 | 0.000 | 0.000 |
| NIG | 0.174 | 0.099 | 0.293 | −0.033 | 0.362 * | 0.628 *** | 0.708 *** | 0.628 *** | 0.580 ** |
|  | 0.309 | 0.567 | 0.082 | 0.850 |  | 0.000 | 0.000 | 0.000 | 0.000 |
| NRA | 0.471 ** | 0.293 | 0.414 ** | 0.094 | 0.265 | 0.563 ** | 0.770 *** | 0.702 *** | 0.759 *** |
|  | 0.004 | 0.083 | 0.012 | 0.587 | 0.119 |  | 0.000 | 0.000 | 0.000 |
| SEN | 0.215 | −0.002 | 0.401 ** | 0.241 | 0.216 | 0.342 * | 0.093 | 0.731 *** | 0.821 **** |
|  | 0.208 | 0.989 | 0.016 | 0.156 | 0.206 | 0.042 |  | 0.000 | 0.000 |
| TAN | 0.319 | 0.228 | 0.658 *** | 0.687 *** | 0.331 * | 0.291 | 0.535 ** | −0.014 | 0.763 *** |
|  | 0.058 | 0.181 | 0.000 | 0.000 | 0.049 | 0.085 | 0.001 |  | 0.000 |
| ZAM | 0.241 | −0.067 | 0.336 * | 0.088 | 0.224 | 0.113 | 0.453 ** | 0.201 | 0.010 |
|  | 0.157 | 0.699 | 0.045 | 0.611 | 0.189 | 0.513 | 0.006 | 0.240 |  |

Note: Strength of correlation: no star (0.00–0.19) very weak; * (0.20–0.39) weak; ** (0.40–0.59) moderate; *** (0.60–0.79) strong; **** (0.80–1.0) very strong. In italic the two-tailed statistical significance.

Table 9 shows that a sectoral injection of one unit in all activities determines an average positive distributive effect on rural household income that is more intense the poorer the household. The increase in the income of the poorest rural households (RU_1) compared with the base year is between 5.4% in the DRC and 66.3% in Zambia. Apart from Kenya, the poorest urban households (UR_1), on average, also benefit from a positive redistributive

effect. In Ethiopia, Tanzania, Zambia, Nigeria, and Senegal, this positive impact is also extended to poor urban households (UR_2) but with reduced levels of intensity.

**Table 9.** Average RDM by household category and country.

|  | DRC | Ethiopia | Kenya | Tanzania | Zambia | Mali | Niger | Nigeria | Senegal |
|---|---|---|---|---|---|---|---|---|---|
| RU_1 | 1.054 | 1.236 | 1.100 | 1.115 | 1.663 | 1.111 | 1.117 | 1.231 | 1.440 |
| RU_2 | 1.067 | 1.187 | 1.098 | 1.100 | 1.718 | 1.115 | 1.117 | 1.191 | 1.345 |
| RU_3 | 1.078 | 1.131 | 1.088 | 1.076 | 1.673 | 1.068 | 1.116 | 1.142 | 1.244 |
| RU_4 | 1.056 | 1.060 | 1.051 | 1.060 | 1.411 | 0.989 | 1.110 | 1.080 | 1.209 |
| RU_5 | 1.085 | 0.917 | 1.011 | 1.030 | 1.093 | 0.910 | 1.052 | 0.982 | 1.093 |
| UR_1 | 1.008 | 1.148 | 0.945 | 1.084 | 1.026 | 1.061 | 1.173 | 1.094 | 1.072 |
| UR_2 | 0.953 | 1.114 | 0.980 | 1.006 | 1.018 | 0.964 | 0.842 | 1.019 | 1.027 |
| UR_3 | 0.936 | 1.027 | 1.000 | 0.986 | 0.944 | 1.015 | 0.926 | 0.988 | 0.990 |
| UR_4 | 0.917 | 0.984 | 0.974 | 0.955 | 0.907 | 0.979 | 1.012 | 0.960 | 0.954 |
| UR_5 | 0.895 | 0.825 | 0.934 | 0.918 | 0.886 | 0.932 | 0.818 | 0.888 | 0.855 |

Figure 1 (Panel 1–9) shows the effect of the exogenous demand injection in the single activities in the sample used by this paper on household income distribution. This impact is country-specific. However, the significant positive income multiplier effect of the key agricultural activities is noticeable in all the sample countries. A one-unit injection in these activities reveals the most favorable bias towards rural household income compared with the exogenous demand injection in other key activities. It is followed by the income multiplier effect produced by the agriculture-related sectors.

In all countries but DRC, the income of the rural households in the two lower quantiles benefits the most from this stimulus. In these countries, poor urban households also benefit from a one-unit demand injection in key agricultural activities. This impact is more significant than that on the urban households in the higher income categories. In fact, these latter are generally characterized by a degressive redistributive effect. However, poor urban households benefit less than poor rural households from the simulated exogenous shock. Therefore, the positive reallocative mechanisms from the key agricultural sectors can promote poverty reduction in rural and urban areas considered separately, but not between rural and urban poor households.

Niger is an exception. In fact, in these countries, the positive multiplier effect on the most deprived urban category of households dominates in the majority of key sectors with the potential to improve, in this way, the rural–urban divide in household poverty. The literature indicates that urban and rural economies are strongly interconnected in Niger. This interconnection is primarily due to the geographic proximity of the smaller towns to the rural areas and the stimulus provided by the agricultural growth to the manufacturing sector and demand for row inputs for agricultural processing in the urban areas [62]. Moreover, economic development is expected to reinforce these linkages. Growth is intensely concentrated in the cities [63]. In contrast with the rest of Sub-Saharan Africa, Niger has a relatively low rate of rural–urban migration and urban population share. On the contrary, the rural population is continuously growing and dominated by subsistence farmers or pastoralists.

The DRC shows another different story. The households in the lower income quantile in rural and urban areas are not the most important beneficiaries of the effect of the simulated scenarios. This situation can be explained, at least partly, by the conflict situation in the country that mainly affects the rural population and smallholder farmers [64]. In addition, the ongoing violence and conflict are generating a large number of internally displaced persons, which has almost reached 6 million people [65]. They represent the

poorest segment of the population. As Cazabat and Yasukawa [66] have shown, people with this status have a limited capacity to contribute to the economy.

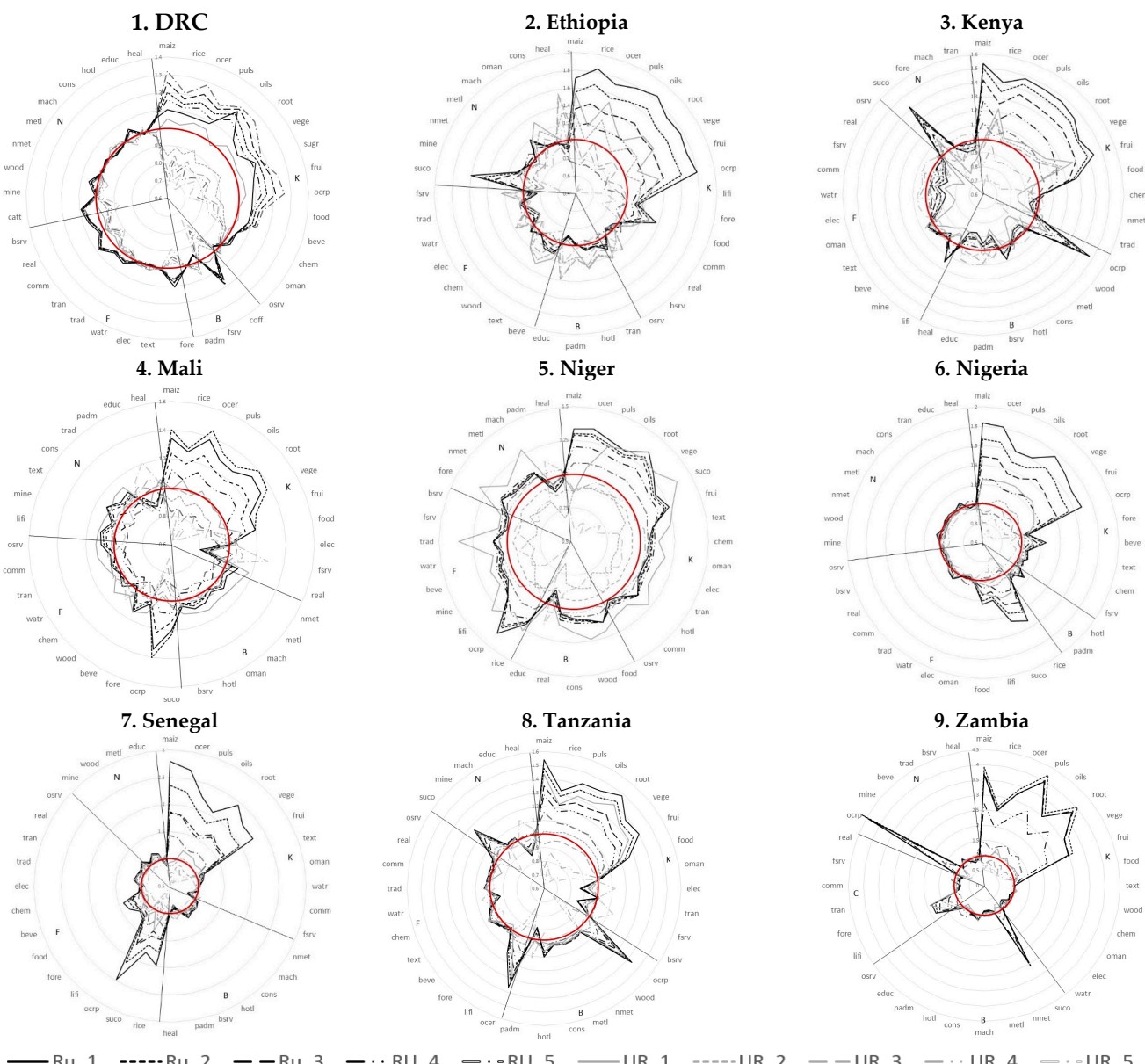

**Figure 1.** RDM by sector, household category and country (the red circle represents the base year).

Another notable piece of evidence from the present analysis concerns the positive redistributive effect, especially on poor rural households, from the injection in agricultural activities despite their level of "keyness". This result confirms the strong relationship between agricultural development and poverty reduction in rural areas highlighted by the literature in Sub-Saharan Africa [9,67]. Furthermore, the paper adds to the literature showing that, in many cases, this positive effect extends to poor urban households, despite the lower levels of intensity.

## 4. Conclusions

As a first step, the paper has identified the critical sectors for economic growth in nine Eastern, Western, and Southern Sub-Saharan African countries by computing the *BL* and *FL* indices for 37 activities. Apart from the countries' specificities, the results confirm the strong integration of agriculture with the economy suggested by the literature and,

therefore, its pivotal role in economic growth in the analyzed countries. The paper adds to this evidence the growth potential of some downstream activities in the agricultural value chain, among which there is the processed food industry in several countries. The situation can be, at least in part, attributed to the gradual evolution of these chains in Africa from informal to more formalized exchanges [68]. This transformation is relevant in the analyzed region due to its potential impact on job creation, increase in agricultural produce value-added, and improvement in dietary diversity and nutritional outcomes [8]. Agrifood value chains can reinforce the positive impact of agriculture on economic growth. From a policy perspective, the conclusion of this paper supports the current focus on enhancing the competitiveness of the African agriculture and agribusiness sectors as an effective approach to creating growth and lessening the dominating poverty phenomena in rural areas [69].

On this last aspect, the *RDM* approach applied in the analysis has shown the strong multiplicative potential effect of a unit exogenous demand injection in the key agricultural sectors on the income of rural households in the lowest quantiles. Therefore, these activities contribute significantly to pro-poor growth in rural areas confirming the current literature on the topic. However, the paper adds to this literature that injection in the key agricultural sectors also impacts the poorest urban household income positively but with a lower intensity. Consequently, the agricultural sector growth contributes to increasing rural–urban income inequality, ceteris paribus. Although the results of the current study should be interpreted carefully in terms of policy implications [4], this latter evidence deserves specific attention. In fact, Sub-Saharan Africa shows the highest rate of urbanization [70], a process associated with growing poverty and inequalities in these areas [71,72]. Despite the beneficial multiplier effect of agriculture growth on rural and urban poverty, to promote pro-poor growth, the effort to boost the sector output and productivity needs to be complemented with policies targeted at the poor urban households, especially in countries where urban areas are not able to absorb the high rates of rural–urban migration. In other words, pro-poor policy priorities should have a component that varies spatially.

This paper used an unconstrained multiplier model based on several limiting assumptions. Among them, this model considers prices as fixed implying that resource factors in the analyzed countries are unlimited or unconstrained. Therefore, the multiplier analysis and *RDM* approach used in this paper allowed to verify the potential effects of exogenous policy shocks directed to key sectors on the income of different households grouped according to income categories, assuming a fixed price environment. Tor these implications to occur, sectoral bottlenecks should be clearly understood and addressed. Among them are the transformation and modernization of agriculture. Broadly speaking, this paper highlights that agriculture in the countries analyzed is less integrated with the upstream sectors than with the downstream sector and shows lower productivity compared with the other economic activities. The great abundance of labor and high rate of poverty in rural areas are important elements that hinder labor-saving technological changes and transitions from traditional farming into commercial and specialized systems for small farmers [73]. Therefore, developing a new agrarian system integrated with the agribusiness sectors seems to represent an important direction to make agriculture an effective contributor to pro-poor growth in Africa.

**Funding:** This research received no external funding.

**Institutional Review Board Statement:** Not applicable.

**Informed Consent Statement:** Not applicable.

**Data Availability Statement:** Data are available at https://www.ifpri.org/project/nexus-project accessed on 4 January 2023.

**Acknowledgments:** I would like to express my very great appreciation to Giovanni Andrea Cornia for his valuable and constructive suggestions during the planning of this study.

**Conflicts of Interest:** The author declares no conflict of interest.

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
