# Peer review of "Economic Connectiveness and Pro-Poor Growth in Sub-Saharan Africa: The Role of Agriculture"

_sustainability, doi:10.3390/su15032026_

Round 1
Reviewer 1 Report
Overall the paper is well-written and well-structured, dealing with an interesting side of research. However, I have a few comments for the authors:
1. The authors have substantially used the word “We, us, and our”. It is advised to avoid such non-academic words and replace them with academic words, such as this research, the current study, etc.
2. The study is lacking literature and Theoretical discussion, the following articles will help the authors to develop these sections.
https://doi.org/10.1111/aswp.12152
https://doi.org/10.5958/2249-7315.2017.00441.5
https://doi.org/10.1007/s11135-013-9885-6
https://doi.org/10.1007/s11135-021-01205-8
3. What is this statement? This is written in “Data and methods” paragraph number 3;
“The current state of the research field should be carefully reviewed and key publications cited. Please highlight controversial and diverging hypotheses when necessary. Finally, briefly mention the main aim of the work and highlight the principal conclusions. As far as possible, please keep the introduction comprehensible to scientists outside your particular field of research. References should be numbered in order of appearance and indicated by a numeral or numerals in square brackets—e.g., [1] or [2,3], or [4–6]. See the end of the document for further details on references.”
4. What sample was used in the current study? How does it collect?
5. The results section is well explained, however, the authors did not write the recommendations based on current findings.
6. Lastly, also add the theoretical and practical implications of the study as well as the limitations of this study.
Author Response
Dear Reviewer,
I appreciate the time and effort that you have dedicated to providing your valuable feedback on my manuscript. I am grateful for your insightful comments on my paper. I have been able to incorporate changes to reflect all the suggestions you have provided to us. In the following, please find my response to your comments.
Comment 1: The authors have substantially used the word “We, us, and our”. It is advised to avoid such non-academic words and replace them with academic words, such as this research, the current study, etc.
Response: The style of the paper has been adjusted as suggested.
Comment 2: The study is lacking literature and Theoretical discussion, the following articles will help the authors to develop these sections.
https://doi.org/10.1111/aswp.12152
https://doi.org/10.5958/2249-7315.2017.00441.5
https://doi.org/10.1007/s11135-013-9885-6
https://doi.org/10.1007/s11135-021-01205-8
Response: The suggested papers were not included because they are not pertinent to our study based on the pioneering work of Rasmussen (1956), Chenery and Watanabe (1958) and Hirschman (1958), contributions mentioned in the paper. Moreover, the papers suggested reffer to a different region.
Comment 3: What is this statement? This is written in “Data and methods” paragraph number 3;
“The current state of the research field should be carefully reviewed and key publications cited. Please highlight controversial and diverging hypotheses when necessary. Finally, briefly mention the main aim of the work and highlight the principal conclusions. As far as possible, please keep the introduction comprehensible to scientists outside your particular field of research. References should be numbered in order of appearance and indicated by a numeral or numerals in square brackets—e.g., [1] or [2,3], or [4–6]. See the end of the document for further details on references.”
Response: The statement has been removed.
Comment 4: What sample was used in the current study? How does it collect?
Response: The section Methodology addresses all these aspects.
Comment 5: The results section is well explained, however, the authors did not write the recommendations based on current findings.
Response: Recommendations are highlighted in the last paragraph of the Conclusions
Comment 6: Lastly, also add the theoretical and practical implications of the study as well as the limitations of this study.
Response: Limitations have been added to the last paragraph of the Conclusions. The paper is not Theoretical. Therefore, it does not add in this direction. However, the policy and research practical implications are indicated in the Conclusions
The reviewed manuscript is in the attachment
Reviewer 2 Report
The manuscript describes a complete analysis of different economic sectors, focused on the role of the agricultural one in relationship with the possibilities of improving its function in the whole economic sector.
The analysis is enough robust and only the following considerations could be considered:
Main remark: Perhaps a deeper discussion would be done to explain the different solutions/possibilities in the way of How to enhance the agricultural sector's principal role.
Minor remarks:
Line 118. IFPRI means???
Line 155, line 281. Capital letter for all agricultural products.
Table 2. DRC means? RU-1, 2, 3 means?
Line 300-302. “ A common feature…..”This statement should be referenced.
Line 325 . GDP means?
Author Response
Dear Reviewer,
I appreciate the time and effort that you have dedicated to providing your valuable feedback on my manuscript. I am grateful for your insightful comments on my paper. I have been able to incorporate changes to reflect all the suggestions you have provided to us. In the following, please find my response to your comments.
Comment 1: Perhaps a deeper discussion would be done to explain the different solutions/possibilities in the way of How to enhance the agricultural sector's principal role.
Response: The indication of solutions and possibilities to enhance the pro-poor role of agriculture are strongly country-specific and to be formulated deserves specific analysis that are outside the scope of the paper, that is, the analysis of the potential role of agriculture in promoting pro-poor growth in rural and urban areas compared to that of other activities.
Comment 2: Line 118. IFPRI means???
Response: The acronym for the International Food Policy Research Institute has been expanded in the Introduction and introduced in the body of the paper after the name in full on its first use.
Comment 3: Line 155, line 281. Capital letter for all agricultural products.
Response: This comment has not been incorporated because the standard IFPRI notation was preferred.
Comment 3: Table 2. DRC means? RU-1, 2, 3 means?
Response: The acronym has been introduced after the name of the country in full on its first use in the text.
RU and UR have been explained in the text.
Comment 4: Line 300-302. “ A common feature…..”This statement should be referenced.
Response: The references, located at the end of the paragraph, have been moved at the end of the indicated sentence.
Comment 5: Line 325 . GDP means?
Response: The acronym has been expanded and then used.
The reviewed paper is in the attachment.
Reviewer 3 Report
Comments and Suggestions for Authors
sustainability-2175138
The article presents an assessment of multiple aspects of Economic connectiveness and pro-poor growth in Sub-Saharan Africa: the dual role of agriculture
This article is well written, well structured, and uses an extended and up-to-date set of references. This post also provides interesting background information on the problem described and a few minor issues that came up during my review:
1. The English grammar and style should be checked throughout the paper.
2. Author need to add clear objectives of the study.
3. What does your article bring to the research field that other papers did not address? I think this must be clearly established to highlight the reader about novelty statement of this article.
4. The figure one can be improved in:
1. Quality and resolution
2. Make sure they are created by the authors ( original) or at least properly cited.
5. Pay more attention to referencing Tables properly
6. Conclusions and recommendations must be clearly related to the results. These relationships should be included in the text.
7. The authors should mention the main limitations of this study at the end of the conclusion section in one paragraph.
8. Follow the sequence of citations.
9. Please improve the manuscript title.
Author Response
Dear reviewer,
I appreciate the time and effort that you have dedicated to providing your valuable feedback on my manuscript. I am grateful for your insightful comments on my paper. I have been able to incorporate changes to reflect all the suggestions you have provided to us. In the following, please find my response to your comments.
Comment 1: The English grammar and style should be checked throughout the paper.
Response: The style of the paper has been adjusted. For example, following the indication of other reviewers, the words “We, us, and our” have been replaced with academic words, such as this research, the current study, etc.
The paper has been submitted to a professional proofreading process.
The style of the Tables was adjusted where possible.
Comment 2: Author need to add clear objectives of the study.
Response: The objective of the paper has been made explicit in the Abstract and in lines 75-78 in the Introduction.
Comment 3: What does your article bring to the research field that other papers did not address? I think this must be clearly established to highlight the reader about novelty statement of this article.
Response: In the Abstract, it has better clarified the objective of the paper that the aspect is still lacking rigorous empirical support.
In the Introduction, it is clarified the novelty of the paper in terms of
- the approach used; and
- focus on households in rural and urban areas and income quantiles.
In the Results and discussion, the new evidence concerning the positive redistributive effect on rural and urban areas and income quantile is identified as a new element provided by the paper to the literature. This aspect is expanded in the Conclusions
Comment 4: The figure one can be improved in:
4.1. Quality and resolution
Response: The quality of Figure 1 has been improved.
4.2. Make sure they are created by the authors (original) or at least properly cited.
Response: The suggested aspects have been verified.
Comment 5: Pay more attention to referencing Tables properly
Response: The source of Table 7 has been improved.
Comment 6: Conclusions and recommendations must be clearly related to the results. These relationships should be included in the text.
Comment 7: The authors should mention the main limitations of this study at the end of the conclusion section in one paragraph.
Response: In the Conclusions, the coherence of the last paragraph with the paper has been better clarified linking this paragraph to the limitations of the analysis in terms of use of and unconstraint model.
Comment 8: Follow the sequence of citations.
Response: Reference n. 65 has been included in the text.
Comment 9: Please improve the manuscript title.
Response: In the title, the word dual has been removed.
The reviewed manuscript is attached.
Reviewer 4 Report
The paper titled Economic connectiveness and pro-poor growth in Sub-Saharan Africa: the dual role of agriculture represents a major contribution to the field of economy and agricultural development in Sub-Saharan Africa. In this paper, the author identified the level of ‘keyness’ of 36 activities in nine Eastern, Western, and Central African countries using the inter-industry linkages analysis, and investigated the income distribution multipliers effects of activities growth across households classified in quantiles in different areas. This topic is very important, original, and relevant in this field of research.
The methodology used is adequate for this type of research.
The author confirmed that economic growth is an essential requirement and often the primary contributor to poverty reduction. Also, the critical sectors for economic growth in Sub-Saharan African countries are identified.
Quotations are relevant, and the references are appropriate, which is very important for further research. The research design is appropriate, and the methods are adequately described. The results are presented adequately and comprehensibly. All figures and tables are well-presented and clear.
The conclusion is supported by the results, and they are consistent with the evidence and arguments presented in the manuscript.
It is not clear whether only one author wrote the manuscript. The abstract (and the whole paper) is written in the first-person plural, and it is stated that there is only one author. Certainly, the manuscript should be reformulated (for example line 10 – This paper instead Our paper; then we investigate; we add; etc.).
Before line 339, after table 8 should be a space.
The article is acceptable for publication in Sustainability after minor revision.
Author Response
Dear Reviewer,
I appreciate the time and effort that you have dedicated to providing your valuable feedback on my manuscript. I am grateful for your insightful comments on my paper. I have been able to incorporate changes to reflect all the suggestions you have provided to us. In the following, please find my response to your comments.
Comment 1: It is not clear whether only one author wrote the manuscript. The abstract (and the whole paper) is written in the first-person plural, and it is stated that there is only one author. Certainly, the manuscript should be reformulated (for example line 10 – This paper instead Our paper; then we investigate; we add; etc.).
Response: The style of the paper has been adjusted. For example, following the indication of other reviewers, the words “We, us, and our” have been replaced with academic words, such as this research, the current study, etc.
Comment 2: Before line 339, after table 8 should be a space.
Response: The space has been added.
The reviewed paper is attached
Round 2
Reviewer 1 Report
The author revised the manuscript substantially following the given comments. However, the manuscript still lacks the literature and Theoretical discussion. I will suggest adding a literature discussion in a separate section.
Author Response
Dear Reviewer,
I want to thank you for the additional comments on my manuscript.
According to the instructions for authors (https://www.mdpi.com/journal/sustainability/instructions), research manuscript sections are expected to be as follows:
- “Introduction: The introduction should briefly place the study in a broad context and highlight why it is important. It should define the purpose of the work and its significance, including specific hypotheses being tested. The current state of the research field should be reviewed carefully and key publications cited. Please highlight controversial and diverging hypotheses when necessary. Finally, briefly mention the main aim of the work and highlight the main conclusions. Keep the introduction comprehensible to scientists working outside the topic of the paper.
- Materials and Methods: They should be described with sufficient detail to allow others to replicate and build on published results. New methods and protocols should be described in detail while well-established methods can be briefly described and appropriately cited. Give the name and version of any software used and make clear whether computer code used is available. Include any pre-registration codes.
- Results: Provide a concise and precise description of the experimental results, their interpretation as well as the experimental conclusions that can be drawn.
- Discussion: Authors should discuss the results and how they can be interpreted in perspective of previous studies and of the working hypotheses. The findings and their implications should be discussed in the broadest context possible and limitations of the work highlighted. Future research directions may also be mentioned. This section may be combined with Results.
- Conclusions: This section is not mandatory but can be added to the manuscript if the discussion is unusually long or complex.
- Patents: This section is not mandatory but may be added if there are patents resulting from the work reported in this manuscript.”
For this reason, I incorporated the relevant literature and theoretical discussion in the different sections, especially in the Introduction and discussion methods. I hope you can approve of this choice.